# A Review of Resistance Mechanisms to Bruton’s Kinase Inhibitors in Chronic Lymphocytic Leukemia

**DOI:** 10.3390/ijms25105246

**Published:** 2024-05-11

**Authors:** Kamil Wiśniewski, Bartosz Puła

**Affiliations:** Department of Hematology, Institute of Hematology and Transfusion Medicine, 02-776 Warsaw, Poland; bpula@ihit.waw.pl

**Keywords:** BTK inhibitors, resistance, ibrutinib, acalabrutinib, zanubrutinib, pirtobrutinib

## Abstract

Bruton’s Tyrosine Kinase (BTK) inhibitors have become one of the most vital drugs in the therapy of chronic lymphocytic leukemia (CLL). Inactivation of BTK disrupts the B-cell antigen receptor (BCR) signaling pathway, which leads to the inhibition of the proliferation and survival of CLL cells. BTK inhibitors (BTKi) are established as leading drugs in the treatment of both treatment-naïve (TN) and relapsed or refractory (R/R) CLL. Furthermore, BTKi demonstrate outstanding efficacy in high-risk CLL, including patients with chromosome 17p deletion, *TP53* mutations, and unmutated status of the immunoglobulin heavy-chain variable region (*IGHV*) gene. Ibrutinib is the first-in-class BTKi which has changed the treatment landscape of CLL. Over the last few years, novel, covalent (acalabrutinib, zanubrutinib), and non-covalent (pirtobrutinib) BTKi have been approved for the treatment of CLL. Unfortunately, continuous therapy with BTKi contributes to the acquisition of secondary resistance leading to clinical relapse. In recent years, it has been demonstrated that the predominant mechanisms of resistance to BTKi are mutations in *BTK* or phospholipase Cγ2 (*PLCG2*). Some differences in the mechanisms of resistance to covalent BTKi have been identified despite their similar mechanism of action. Moreover, novel mutations resulting in resistance to non-covalent BTKi have been recently suggested. This article summarizes the clinical efficacy and the latest data regarding resistance to all of the registered BTKi.

## 1. Introduction

Chronic lymphocytic leukemia (CLL) is an indolent lymphoproliferative malignancy characterized by a progressive accumulation of monoclonal and dysfunctional B lymphocytes. The elderly population comprise the majority of CLL patients, with a median age at diagnosis of 72 years [1,2,3,4]. CLL is one of the most common hematological malignancies, being responsible for nearly 40% of all leukemia cases in adults [5]. The B-cell antigen receptor (BCR) signaling plays a vital role in leukemogenesis of CLL. Aberrant chronic and tonic BCR signaling supports the proliferation and survival of CLL cells. Bruton’s Tyrosine Kinase (BTK) constitutes an essential part of the BCR signaling pathway. Therefore, BTK has become one of the most valuable targets for novel therapies in CLL and other B-cell malignancies [6,7]. Currently, there are two classes of BTK inhibitors (BTKi) available: covalent and non-covalent. Ibrutinib is the first covalent BTKi (cBTKi) registered for the treatment of CLL. It has changed the standard of care for CLL patients, enabling them to achieve historically high rates of response and survival [8,9]. The second generation cBTKi (acalabrutinib and zanubrutinib) manifest even greater selectivity toward BTK, allowing them to reduce the off-target toxicity [10,11,12,13]. Finally, pirtobrutinib, a non-covalent BTKi (ncBTKi), represents a novel class of BTKi developed to improve effectiveness and overcome acquired resistance to cBTKi [14,15].

Despite the durable responses of BTKi, many patients suffer from secondary resistance and disease progression [16,17,18,19,20]. In addition, some of them develop Richter’s transformation (RT) with an extremely poor prognosis [21,22,23,24,25]. Unfortunately, the mechanism of development of the resistance to BTKi is not fully understood. It has been widely accepted that the main mechanism of resistance to cBTKi involves mutations in *BTK* and, to a lesser extent, phospholipase Cγ2 (*PLCG2*) [22,23,26,27,28]. Nonetheless, in recent years, some studies have provided conflicting conclusions, as these mutations were not detected in a significant rate of patients relapsing on cBTKi [19,29]. In addition, there are new data regarding potential mechanisms of secondary resistance to ncBTKi.

The objective of the present article is to briefly review all the registered BTKi in terms of their clinical efficacy. Furthermore, the authors discuss the latest reports concerning the possible mechanisms of resistance development in patients with CLL treated with BTKi.

## 2. Ibrutinib

Ibrutinib is the first-in-class BTK inhibitor that has revolutionized the therapy of CLL. Ibrutinib covalently binds to the cysteine 481 (C481) residue of the BTK active site, irreversibly restraining the BCR signaling pathway, which contributes to reduced activation, migration, proliferation, and survival of CLL cells. Treatment with ibrutinib has significantly improved overall survival (OS) and the quality of life of CLL patients [8,9,30,31,32]. The remarkable efficacy of ibrutinib in both first-line and relapsed or refractory CLL has been shown in several studies. In the initial phase of the 1b/2 PCYC-1102 study and the PCYC-1103 extension study, the overall rate response (ORR) reached 89% with close results of 87% and 89% in treatment-naïve (TN) and relapsed/refractory (R/R) patients, respectively. With a follow-up of 8 years, the median progression-free survival (PFS) was not reached (NR) within TN patients and was 52 months in an R/R setting. The estimated 7-year PFS and OS rates for patients receiving ibrutinib first-line treatment were 83% and 84%, respectively, while for patients with R/R CLL, they were 34% and 55%, respectively [8]. The RESONATE trial was the first randomized phase 3 study that compared ibrutinib to ofatumumab in patients with R/R CLL [9]. At a median follow-up of 65.3 months, the median PFS was significantly longer in the ibrutinib vs. ofatumumab arm (44.1 vs. 8.0 months). The ORR in the ibrutinib arm was 91%, including an 11% rate of complete response/complete response with incomplete bone marrow recovery (CR/CRi) [31]. In the RESONATE-2 trial, ibrutinib significantly improved response rates, OS, and PFS in comparison with chlorambucil in patients with previously untreated CLL [32]. With a follow-up of 8 years, the median PFS was NR for patients treated with ibrutinib and 15 months for patients receiving chlorambucil [33]. Most importantly, ibrutinib was very effective in a population with historically inferior survival, including patients with del(17p), *TP53* mutations and the unmutated immunoglobulin heavy-chain variable region (*IGHV*) gene [8,9,30,31,32].

Although ibrutinib is a vital drug for CLL patients, many of them experience disease progression (PD) at some point in treatment. For this reason, many studies have been conducted to explore and identify potential factors of resistance to ibrutinib [16,22,23,26,27,28,29,34]. One of the first data concerning BTKi resistance came from the 2014 year when Furman et al. reported a case of a patient with CLL progression on ibrutinib [16]. The authors performed RNA sequencing of the blood sample collected at the time of PD. They detected a mutation in the *BTK*, resulting in the substitution of serine for cysteine at residue 481 (C481S). It has been demonstrated that the C481S mutation disrupts ibrutinib binding to BTK, resulting in a reversible connection instead of an irreversible one. This exchange only leads to a transient inhibition of BTK by ibrutinib. The C481S mutation had not been present before ibrutinib administration. In Woyach et al.’s study, the same mutation was found in 83% (5/6) of patients relapsing on ibrutinib. Additionally, the authors have identified three distinct mutations in *PLCG2* (S707Y, R665W, L845F) in two patients (33%). All of the mutations in *PLCG2* manifest a potential gain of function that results in BCR signaling activity independently from BTK inhibition. Similarly, in all of the cases, *BTK* and *PLCG2* mutations had not been detected before the initiation of ibrutinib therapy. Some patients harbored mutations in both BTK and PLCG2 enzymes [26]. These findings have been confirmed in several trials where the frequency of *BTK* and *PLCG2* mutations was 80–100% at disease progression [22,23,27]. Moreover, additional mutations in *BTK* (C481R, C481F, C481A, C481Y, T316A, T474, L528) and *PLCG2* (S707P, S707F, P664S, D933G, D933H, D993Y, D1140N, D1140G, M1141K, M1141R, E1139G) have been described [16,21,22,23,26,27,28,29,35]. Therefore, mutations in *BTK* and *PLCG2* are considered to be the predominant mechanism of ibrutinib resistance in patients with CLL. Furthermore, in several studies the detection of *BTK* and/or *PLCG2* mutations preceded the clinical progression of CLL by 8–10 months [23,27,36,37]. In a real-life study carried out by the French Innovative Leukemia Organization (FILO) group, the existence of mutation in *BTK* was significantly related to subsequent PD [36]. In Ahn et al.’s study, the presence of *BTK* and/or *PLCG2* mutations was manifested in a median time of 8 months before CLL progression. with an increasing variant allele frequency (VAF) [23].

Recently, Wiestner et al. conducted a large analysis of *BTK* and *PLCG2* mutation in 388 patients treated with ibrutinib in clinical trials. The study included patients who received ibrutinib in the first (RESONATE-2, iLLUMINATE, and NCT01500733) and subsequent lines of therapy (RESONATE and RESONATE-17). The analysis showed that *BTK* mutations are significantly more frequent in patients with R/R CLL as compared to TN CLL treated with ibrutinib (30% vs. 3%). Similarly, mutations of *PLCG2* were detected at a higher rate in the R/R setting (7% vs. 2%). The median time to *BTK* mutation detection was significantly longer in patients treated with ibrutinib in first-line therapy (NR vs. 58 months). Additionally, the presence of del(17p) or *TP53* mutations was associated with a higher risk of acquisition of *BTK* mutations [38].

It remains unexplained why CLL patients with low levels of *BTK* or *PLCG2* mutations experience disease progression on BTKi. It is well known that the progression of CLL can have a variable clinical presentation. Particularly, in some patients it manifests mainly as progressive lymphadenopathy. However, the majority of studies have only assessed peripheral blood samples and do not include lymph node samples. According to the data collected so far, BTKi resistance is polyclonal. Thus, it is possible that particular clones occur at a different level depending on the site of resistance. In Woyach et al.’s study, disease progression was found only in the lymph nodes in six of seven patients with a low VAF of *BTK* mutations (<10%) [27]. Therefore, it has been suggested that additional testing of lymph node samples may be necessary to gain a full understanding of resistance to BTKi [34]. Finally, clones with low VAF *BTK* or *PLCG2* mutations may coexist with alternative aberrations, contributing to secondary resistance development.

Nevertheless, there is a significant group of patients who do not harbor mutations, neither in *BTK* nor in *PLCG2*. In a retrospective real-world observational study conducted by the European Research Initiative on CLL (ERIC), the mutations in *BTK* and/or *PLCG2* were not present in 35% of patients at the time of progression [29]. What is more, it is unclear if a single mutation in *BTK* or *PLCG2* is sufficient to develop resistance to ibrutinib. Furthermore, in numerous patients, the PD was documented along with a low VAF of *BTK* or *PLCG2* mutations (<30%), including cases with VAF < 10% [23,27,34]. Surprising results were obtained from the analysis of the ALPINE study where mutations of *BTK* or *PLCG2* were detected in only 14% (4/28) of patients relapsing on ibrutinib [19].

Therefore, numerous studies have been performed and some potential mechanisms of resistance to ibrutinib have been suggested [21,29,33,39,40]. Burger et al. observed del(8p) with additional driver mutations in other genes (*EP300*, *MLL2* and *EIF2*) in three patients with resistance to ibrutinib therapy. Chromosome 8p deletion results in haploinsufficiency of tumor necrosis factor-related apoptosis-inducing ligand-receptor (TRAIL-R), which may lead to progression on ibrutinib [33]. Bonfiglio et al. confirmed the enrichment of del(8p) in 11 relapsing patients, but most of them also harbored *BTK*/*PLCG2* mutations [29]. In the same study, the authors described the presence of mutations in the early growth response 2 (*EGR2*) gene in patients with PD coexisting with *BTK* mutations. Furthermore, the authors have identified other recurrent gene mutations in *ASXL1*, *BRAF*, *IKZF3*, *KRAS*, *MED12*, *MGA*, *RPS15*, *SPEN*, and *ZFN292*. Nonetheless, only two of them (*BRAF* and *IKZF3*) were more common in relapsing patients.

## 3. Acalabrutinib

Acalabrutinib is a novel, second-generation BTK inhibitor. Similarly to ibrutinib, acalabrutinib binds covalently and irreversibly to C481 within the ATP-binding site of BTK. However, acalabrutinib demonstrated higher potency and selectivity for BTK in comparison with ibrutinib, which reduces the off-target toxicity [10,41,42]. In clinical trials, acalabrutinib has shown similar effectivity to ibrutinib with a favorable safety profile. It has been shown that acalabrutinib induces a significantly lower rate of cardiac events compared to ibrutinib. In the ELEVETE-TN trial, acalabrutinib with or without obinutuzumab demonstrated remarkable effectiveness in frontline therapy for CLL, including patients with high-risk genetic features. With a median follow-up of 74.5 months, median PFS was significantly longer with acalabrutinib monotherapy or acalabrutinib-obinutuzumab as compared to obinutuzumab-chlorambucil (NR in both acalabrutinib arms vs. 27.8 months) [43]. In the ASCEND study, patients were treated with acalabrutinib or the investigator’s choice (idelalisib-rituximab [IR] or bendamustine-rituximab [BR]). Acalabrutinib manifested significantly improved PFS as opposed to the investigator’s chosen therapy [11]. In a final analysis (median follow-up of approximately 4 years), a median PFS was NR for patients receiving acalabrutinib and was 16.8 months for patients treated with IR/BR [44]. Finally, the ELEVETE-RR trial was a head-to-head study comparing acalabrutinib against ibrutinib in R/R CLL [45]. Acalabrutinib showed comparable efficacy to ibrutinib with a more favorable safety profile. Patients treated with acalabrutinib experienced fewer cardiovascular events, in particular atrial fibrillation, hypertension, and hemorrhage [46,47].

Recent data indicate that secondary resistance to acalabrutinib is also mainly related to mutations in *BTK* [18,48,49,50]. In Woyach et al.’s study, mutations in *BTK* (C481S, C481R and C481Y) were detected in 11 of 16 relapsing patients (69%). Notably, *BTK* C481S was the predominant mutation of *BTK*, detected in 10 patients. Two patients were identified with two different mutations in *BTK* (*BTK* C481S + *BTK* T474I and *BTK* C481S + *BTK* C481R). Additionally, two patients showed a coincidence of *BTK* C481S and *PLCG2* mutations. Mutations were detected at a median time of 31.6 months from the initiation of acalabrutinib. The median time from the identification of mutation to progression was 12 months, and it did not vary between TN and R/R CLL patients [18]. Similar conclusions were reached by Sun et al. in the mutational analysis of patients treated with acalabrutinib in the randomized phase 2 study (NCT02337829) [48]. Mutations in *BTK* and *PLCG2* were detected in 6/20 (30%) and 4/20 (20%) of patients, respectively. *BTK* C481S was the most common *BTK* mutation (11/20, 55%). *BTK* T474I mutation was found in four patients and always coexisted with *BTK* C481S. In the ELEVETE-RR trial, *BTK* and *PLCG2* mutations were also the most common mutations identified in patients treated with ibrutinib and acalabrutinib. However, there were some differences in mutation frequency between both arms: *BTK* mutations were found in 31 (66%) of relapsing patients treated with acalabrutinib and 11 (37%) in the ibrutinib group. Mutations in *PLCG2* were more common in ibrutinib (6/30, 20%) compared with the acalabrutinib group (3/47, 6%). Moreover, a novel *TK* E41V mutation (VAF 16%) was detected in patients with PD treated with acalabrutinib [49].

## 4. Zanubrutinib

Zanubrutinib is another second-generation covalent BTK inhibitor [51]. It is highly effective in both TN and R/R patients with CLL. The efficacy of zanubrutinib in previously untreated CLL was compared with BR therapy in the SEQUOIA trial. Patients treated with zanubrutinib achieved significantly longer PFS than patients in the BR arm [12]. The estimated 42-month PFS rate reached 82.4% in the zanubrutinib group and 50.0% in the BR group [52]. In addition, the SEQUOIA study included a nonrandomized cohort of “high-risk” CLL patients with del(17p) treated with zanubrutinib (Arm C). The ORR was 94.5%, including a 3.7% rate of CR/CRi. The estimated 18-month PFS and OS rates were 88.6% and 95.5%, respectively [53]. In the ALPINE trial zanubrutinib was found to be superior to ibrutinib with regard to PFS and ORR in patients with R/R CLL. The ORR in the zanubrutinib and ibrutinib group was 86.2% and 75.7%, respectively [13]. In the latest follow-up analysis, the PFS rate reached 65.8% in the zanubrutinib arm compared to 54.3% in the ibrutinib arm. Furthermore, PFS benefit was confirmed across all key subgroups, including the high-risk population (patients with 17p deletion/*TP53* mutation) [54]. In both trials, zanubrutinib demonstrated a better safety profile as compared to BR or ibrutinib therapy [12,13].

Despite the fact that zanubrutinib belongs to covalent BTKi, it has been shown that it may be characterized by different mechanisms of resistance than ibrutinib and acalabrutinib. In Blombery et al.’s study, the authors found that patients progressing on zanubrutinib were significantly more likely to acquire the kinase-dead *BTK* Leu528Trp (L528W) mutation as opposed to patients who received ibrutinib (54% vs. 4%). At the same time, it should be emphasized that *BTK* C481S mutations were found in 77% of patients treated with zanubrutinib and in all patients in the ibrutinib group [55]. In another study, a mutation of *BTK* L528W was detected in all patients with disease progression treated with zanubrutinib. Notably, all of those patients had coexisting *BTK* Cys481 mutations, though with a lower VAF than the *BTK* L528W (9.1% vs. 34.9%) [56]. Nevertheless, in the ALPINE trial, only five patients with PD on zanubrutinib (5/24) acquired mutations in *BTK* (C481, n = 4, L528W, n = 2, A428D, n = 1). None of the patients manifested a mutation of *PLCG2*. In the ALPINE study, the majority of patients treated with both zanubrutinib and ibrutinib had no detectable mutations of *BTK* or *PLCG2* (82.6%). On the basis of the obtained data, the authors have suggested that mutations of *BTK* and/or *PLCG2* may not be the predominant mechanism of resistance to cBTK [19]. In Zhu et al.’s study, *BTK* C481S/R mutations were identified in five out of eight patients with resistance to zanubrutinib. Only one patient harbored a *BTK* L528W mutation. Moreover, *TP53*, *EGR2*, *NOTCH1*, and SF3B1, along with *BTK* C481, were found to be the driving clones selected during zanubrutinib resistance [57].

## 5. Pirtobrutinib

Pirtobrutinib is a potent, highly selective ncBTKi that enables it to break through resistance to cBTKi. Due to its unique structure, pirtobrutinib reversibly links to the ATP binding site of BTK without the involvement of C481 [58]. Therefore, pirtobrutinib remains active in both the C481-mutated and wild-type BTK, overcoming the major mechanism of resistance to cBTKi. The first data on the efficacy of pirtobrutinib in the treatment of CLL came from the phase 1/2 BRUIN trial [14,15]. The updated results of the study have recently been published. To date, the study has included 282 heavily pretreated patients with a median of 4 prior lines of therapy. All of the patients were treated previously with cBTKi. The ORR was 72% and 82%, including partial response with lymphocytosis (PR-L). The median PFS was 19.4 months (median follow-up: 27.5 months) and the median OS was NR (median follow-up of 29.3 months). The study included 128 (45%) “double-refractory” patients previously treated with both BTKi and a B-cell lymphoma 2 (BCL2) inhibitor. In this group of patients with a particularly poor prognosis, the ORR, including PR-L, was 80%. Median PFS was 15.9 months (median follow-up of 15.9 months) [56]. Pirtobrutinib demonstrates a favorable safety profile with a treatment discontinuation rate of only 2% [59,60]. Based on the results from the BRUIN trial, pirtobrutinib has become the first registered ncBTKi for the treatment of CLL.

Unfortunately, despite a high response rate, a subset of patients treated with pirtobrutinib also developed secondary resistance. The first reports on mechanisms of resistance to pirtobrutinib come from the BRUIN study. The study revealed that non-C481 *BTK* mutations (V416L, A428D, M437R, T474I, and L528W) were harbored in seven of nine patients with PD. All these mutations were grouped within the kinase domain of BTK and seemed to limit pirtobrutinib connection with the ATP binding pocket. Non-C481 *BTK* mutations were detected in both patients with baseline wild-type and C481-mutated *BTK*. It is noteworthy that in patients in whom *PLCG2* mutations were detected at baseline, clones with these mutations survived treatment with pirtobrutinib and were present at the moment of progression [61]. In Qi et al.’s study, the investigators used in vitro pirtobrutinib-resistance lines of REC-1 cells to detect mutations in *BTK* and *PLCG2*. The authors detected acquired mutations of *BTK* (A428D) and *PLCG2* (R727L, S1079R) [62]. In another study, Naeem et al. performed whole-exome sequencing of two patients progressing on pirtobrutinib. In the first patient, three mutations were detected within *BTK*: T474I, M477I and T474L, the first two being novel. The other patient manifested the *BTK* L528W mutation at the time of progression [63]. *BTK* L528W mutations had been previously reported in patients relapsing on zanubrutinib [55]. In the same study, the authors described two cases of patients with detectable L528W mutations who were treated with pirtobrutinib after treatment failure with zanubrutinib. In the first case, a rapid increase in the *BTK* L528W clone was observed during treatment with pirtobrutinib until disease progression. The other patient, who initially carried a completely clonal *BTK* L528W mutation, only achieved disease stabilization during treatment with pirtobrutinib, followed by progression. These data indicate that *BTK* L528W contributes to secondary resistance to BTKi as well potentially causing cross-resistance between cBTKi and ncBTKi. Importantly, to date, a *BTK* L528W mutation has not been identified in patients progressing on acalabrutinib. Recently, the results from the BRUIN study, involving 86 patients relapsing on pirtobrutinib, have been published. In total, 52 acquired *BTK* mutations were found in 44% (38/86) of patients, particularly gatekeeper mutations (T474I/F/S/L/Y, 26% of patients), kinase impaired (L528W, 16% of patients), and C481S/R/Y (4% of patients). Non-BTK mutations (n = 83) were detected in 52% (45/86) patients, including *TP53* (14%), *PLCG2* (7%), *PIK3CA* (7%), and *BCL2* (3%) mutations [64]. Mutations of *BTK* and *PLCG2* genes detected in patients treated with BTKi are shown in Table 1.

## 6. Overcoming Resistance to BTKi

Venetoclax is the first approved B-cell lymphoma two inhibitor (BCL2i) for the treatment of CLL. It shows remarkable efficacy in the therapy of both TN and R/R CLL patients [65,66,67]. Currently, venetoclax, together with BTKi, constitute the most important drugs used in CLL treatment. Furthermore, it demonstrates very high activity in CLL patients after BTKi failure [68,69]. The results of a large retrospective study involving 184 patients who discontinued cBTKi and received venetoclax-based treatment were recently published. The ORR was 78.0%, including 43.3% CR and 34.6% PR. It is worth noting that patients treated with venetoclax in the 2nd and 3rd line of treatment achieved a very similar response rate (84.1% and 78.3%) and median PFS (43.2 and 44.1 months) [68]. In Jones et al.’s study, the ORR of patients relapsing on ibrutinib treated with venetoclax was 65% with a median PFS of 23.5 months [69]. Moreover, venetoclax demonstrates promising results in patients with ncBTKi treatment failure. In the retrospective study conducted by Thompson et al., venetoclax yielded an ORR of 70% in patients after ncBTKI discontinuation [70].

Due to the upregulation of phosphoinositide 3-kinase (PI3K) in patients with resistance to ibrutinib, Pi3K inhibitors were expected to be effective in that population [71]. Unfortunately, in one of the first studies, the ORR was only 28% (0% of CR) in patients treated with idelalisib after ibrutinib failure [72]. In another study, Pi3K inhibitors showed a promising ORR of 46.9% in BTKi/BCL2 double-refractory patients. However, remissions were not durable, and the median PFS reached only 5 months [73]. Thus, PI3K inhibitors do not appear to be an optimal treatment option in patients with resistance to BTKi.

Currently, one of the most revolutionary approaches to overcoming BTKi resistance is the potential use of BTK degraders. These drugs, developed by means of the novel proteolysis-targeting chimera (PROTAC) technology, enable the degradation of both mutated and wild-type BTK. PROTACs constitute specialized molecules consisting of two linked domains, one that binds to the targeted protein and the other to E3 ubiquitin ligase. Such binding results in the degradation of the targeted protein by the proteasome [74,75,76]. Several BTK degraders, including NX-2127, NX-5948, BGB-16673, ABBV-101, and AC676, were recently under investigation in CLL patients [77,78,79,80,81]. One of the most advanced research projects is concerned with the NX-2127 molecule. The first-in-human phase I trial shows that NX-2127 induces a mean BTK degradation of 86%. The study enrolled 17 heavily pretreated CLL patients, including 76.5% BTKi/BCL2 double-refractory patients. The following *BTK* mutations were detected: C481 (29%), L528 (29%), T474 (14%), and V416 (7%). The ORR reached 33% with a positive trend to increase over time. Importantly, a clinical response was also achieved in patients previously treated with ncBTKi [77].

Finally, chimeric antigen receptor (CAR) T-cell therapy represents the most novel strategy that is being investigated in R/R CLL [82,83,84,85,86]. The first studies have provided promising results on the effectiveness of CD19-directed CAR-T in CLL [70,82,83,84,85]. In Turtle et al.’s study, 17 of 24 patients achieved remission 4 weeks after CAR-T infusion (71% of ORR). Notably, the majority of patients had high-risk cytogenetic features (complex karyotype and/or 17p deletion) [82]. In the Thompson et al. analysis, CAR-T therapy showed tremendous efficacy with an ORR of 86% [70]. TRANSCEND CLL 004 is the first trial of lisocabtagene maraleucel (liso-cel) in patients with R/R CLL. The study enrolled a total of 127 heavily pretreated CLL patients with a median of 5 previous lines of treatment, including BTKi therapy. Additionally, 83% of patients had a high cytogenetic risk. A single infusion of liso-cel produced deep and durable responses in this challenging patient population [84]. With a follow-up of 23.5 months, the ORR was 48%, and the median duration of response (DOR) was 35 months. The CR/CRi rate was 19%, and the median DOR of patients with CR/Cri was NR. The unmeasurable residual disease (uMRD) rate in blood and bone marrow was 64% and 60%, respectively. The median duration of response was NR for patients with CR/Cri [85]. Based on the data from TRANSCEND CLL 004, liso-cel became the first and only approved CAR-T therapy in patients with R/R CLL.

## 7. Conclusions

The development of targeted therapies has greatly changed the landscape of treatment for CLL. Currently, BTKi are the most important group of drugs for all CLL patients. In recent years, novel and even safer and more effective BTKi have been introduced. However, secondary resistance is an extremely important clinical problem in patients treated with both covalent and non-covalent BTKi. Recently, several mutations that may be responsible for resistance to cBTK have been identified, mainly at the c481 position of the BTK. Nevertheless, the exact mechanism of resistance acquisition still remains unclarified. Many patients treated with pirtobrutinib experience disease progression despite the drug overcoming resistance to cBTK. In addition, particular mutations can induce cross-resistance between BTKi, including non-covalent ones. Therefore, gaining knowledge in this area is essential to optimal sequencing of these agents and improving patient outcomes. Last but not least, the treatment of patients who are refractory to all available BTKi seems to be the greatest clinical challenge. Further studies are needed to better understand the mechanism of resistance to BTKi and to identify the still unknown causes of this phenomenon.

## Figures and Tables

**Table 1 ijms-25-05246-t001:** Mutations of *BTK* and *PLCG2* genes in patients treated with Bruton’s Tyrosine Kinase inhibitors.

Study	No. of Patients with PD	CLL Status	Drug	BTK and PLCG2 Mutations	VAF, %	Reference
Woyach et al., 2014	6	R/R	Ibrutinib	BTK (5/6): C481S PLCG 2 (2/6): L845F, R665W, S707Y	nd	[26]
Maddock et al., 2017	40	R/R	Ibrutinib	BTK (37/40): C481S (34/37), C481F, C481A, C481R, PLCG 2 (7/40): L845F, R665W, S707Y, S707P, S707F, L845/846del, D993Y	0.2–100.0 3.4–44.4	[27]
Ahn et al., 2017	13	TN and R/R	Ibrutinib	BTK (7/13): *C*481*S (7/7)*, *C*481*F* PLCG 2 (7/13): L845F, R665W, S707Y, P664S, 6NT del	1.6–78.2 0.1–18.3	[23]
Kanagal-Shamanna et al., 2019	23	R/R	Ibrutinib	BTL (16/23): C481S, C481F, C481Y, C481R, V537I PLCG2: 0/23	1.0–91.0	[39]
Bonfiglio et al., 2023	49	TN and R/R	Ibrutinib	BTK (24/49): C481R, *C*481*S*, *C*481*Y* PLCG2 (14/49): D1140N, D993Y, M1141K, D993G, M1141R, L848R, D993H, L845F, R665W, S 707F, E1139G, D1140G	1.8–79.5 1.7–32.7	[29]
Woyach et al., 2019	24	R/R	Ibrutinib	BTK (11/24): C481S, C481Y, C481R, L528W, A428D PLCG2 (6/24)	5.8 (median)	[49]
	46	R/R	Acalabrutinib	BTK (31/46): C481S, C481Y, C481R, E41V PLCG2 (2/46)	5.7 (median)	
Woyach et al., 2019	16	TN and R/R	Acalabrutinib	BTK (11/16): C481S (10/11), C481R, C481Y, T474I PLCG2 2/16	nd <3	[18]
Sun et al., 2023	20	TN and R/R	Acalabrutinib	BTK 6/20: C481S 6/6, T474I PLCG 4/20	2.0–46.0 1.0–8.0	[48]
Blombery et al., 2022	24	R/R	Ibrutinib	BTK (24/24): C481S (24/24), C481A, C481F, C418T, T474I, L528W	1.0–79.0	[55]
	13	R/R	Zanubrutinib	BTK (10/13): C481S (10/13), L528W (7/10), C418T	2.0–87.0	
Brown et al., 2023	28	R/R	Ibrutinib	BTK 3/28: C481S (3/3), D43H PLCG2 (2/28)	0.5–8.0	[19]
	24	R/R	Zanubrutinib	BTK (5/24): C481S (4/5), C481Y, C481F, C481R, L528W, A428D PLCG2 (0/9)	1.0–74.0	
Handunnetti et al., 2019	4	TN and R/R	Zanubrutinib	BTK (4/4): C481 (4/4), L528W (4/4)	L528W: 34.9 (median) C418: 9.1 (median)	[56]
Wang et al., 2022	9	R/R	Pirtobrutinib	BTK (7/9): V416L, M437R, T474I, L528W PLCG2 (0/9)	nd	[61]
Brown et al., 2023	86	R/R	Pirtobrutinib	BTK (38/86): T474I, T474F, T474S, T474L, T474Y, L528W, C481S, C481R, C481Y PLCG2 (6/86)	nd	[64]

Abbreviations: PD, progressive disease; CLL, chronic lymphocytic leukemia; VAF, variant allele frequency; R/R, relapsed/refractory; TN, treatment-naïve; BTK, Bruton’s Tyrosine Kinase; PLCG2, phospholipase Cγ2; nd, no data.

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
