# Peer review of "A Review of Resistance Mechanisms to Bruton’s Kinase Inhibitors in Chronic Lymphocytic Leukemia"

_ijms, 2024, doi:10.3390/ijms25105246_

Round 1

Reviewer 1 Report

Comments and Suggestions for Authors

This is a well-written review of resistance mechanism to BTK inhibitor in CLL where the authors walk the reader through resistance to the different ibrutinib drugs on the marked.

Although the focus (and manuscript title) is about  BTK resistance, there should be some consideration about other precision medicine drugs as well. Novel treatment agents are today often used in competition to ibrutinib and this should be discussed along-side with ibrutinib. Thus, the text (e.g. a new paragraph prior to the "Conclusion" or the "Conclusion") could preferentially be expanded to include a perspective to other common novel treatments as e.g. the BCL2 inhibitor venetoclax.

Minor comment:

Section2. Ibrutinib:

What is the meaning of this sentence: "With a follow-up of 8 years, a median progression free survival was not reached with treatment naïve and was 52 months in R/R setting.

How can a median not be reached? Do the authors mean the median of the whole cohort? Please specify.

Reviewer 2 Report

Comments and Suggestions for Authors

This review provides a thorough and scientifically sound analysis of the resistance mechanisms to Bruton’s kinase inhibitors in Chronic Lymphocytic Leukemia (CLL). All recent contributions in the field are meticulously reported and discussed.

I have some minor comments and suggestions:

- The role of BTK degraders as a potentially effective treatment for patients developing BTK mutations should be discussed. Notably, the seminal paper by Montoya et al., published in Science, demonstrating that BTK degraders address drug resistance, warrants inclusion and commentary.

- Another crucial point that requires clarification is the prevalence of BTK mutations in patients receiving BTKis upfront or in the context of relapsed/refractory (R/R) disease. The meta-analysis by Wienster, presented in abstract form at the 2020 ASH meeting (Abstract 2225), aids in elucidating this aspect.

- Results presented at the recent IWCLL 2023 meeting suggest a compartmental effect when assessing BKT mutations. Do the authors have similar information? If so, I would appreciate a brief commentary on this aspect, as it appears significant.

Author Response

We appreciate your time spent on reviewing this manuscript. Please find some clarifications we made below with regards to your valuable comments.

Comments 1: The role of BTK degraders as a potentially effective treatment for patients developing BTK mutations should be discussed. Notably, the seminal paper by Montoya et al., published in Science, demonstrating that BTK degraders address drug resistance, warrants inclusion and commentary.

Response 1: We fully agree with this comment. A new paragraph (7. Overcoming resistance to BTKi) has been added to the article, describing novel treatment agents in R/R CLL, including BTK degraders.

Comments 2: Another crucial point that requires clarification is the prevalence of BTK mutations in patients receiving BTKis upfront or in the context of relapsed/refractory (R/R) disease. The meta-analysis by Wienster, presented in abstract form at the 2020 ASH meeting (Abstract 2225), aids in elucidating this aspect.

Response 2: Thank you for pointing this out. The results of the meta-analysis by Wienster has been added in the Section 2 (Ibrutinib).

Comments 3: Results presented at the recent IWCLL 2023 meeting suggest a compartmental effect when assessing BKT mutations. Do the authors have similar information? If so, I would appreciate a brief commentary on this aspect, as it appears significant.

Response 3: Thank you for this valuable comment. The topic on the possible impact of the compartmental effect on the assessment of BTK resistance has been added in the Section 2 (Ibrutinib).

In addition, the reference list has been revised in line with the changes made.